# Kindlin-2 inhibits TNF/NF-κB-Caspase 8 pathway in hepatocytes to maintain liver development and function

**Huanqing Gao[1†], Yiming Zhong[1†], Liang Zhou[2†], Sixiong Lin[1,3†], Xiaoting Hou[1†], Zhen Ding[1], Yan Li[4], Qing Yao[1], Huiling Cao[1], Xuenong Zou[3], Di Chen[5], Xiaochun Bai[6*], Guozhi Xiao[1*]**

[1]Department of Biochemistry, School of Medicine, Guangdong Provincial Key Laboratory of Cell Microenvironment and Disease Research, Shenzhen Key Laboratory of Cell Microenvironment, Southern University of Science and Technology, Shenzhen, China; [2]Guangdong Province Key Laboratory of Pharmaceutical Functional Genes, MOE Key Laboratory of Gene Function and Regulation, State Key Laboratory of Biocontrol, School of Life Sciences, Sun Yat-sen University, Guangzhou, China; [3]Guangdong Provincial Key Laboratory of Orthopedics and Traumatology, Department of Spinal Surgery, The First Affiliated Hospital of Sun Yat-sen University, Guangzhou, China; [4]Department of Biology, Southern University of Science and Technology, Shenzhen, China; [5]Research Center for Human Tissues and Organs Degeneration, Shenzhen Institutes of Advanced Technology, Chinese Academy of Sciences, Shenzhen, China; [6]Provincial Key Laboratory of Bone and Joint Degeneration Diseases, Department of Cell Biology, School of Basic Medical Sciences, Southern Medical University, Guangzhou, China

**\*For correspondence:**
baixc15@smu.edu.cn (XB);
xiaogz@sustech.edu.cn (GX)

†These authors contributed equally to this work

**Abstract** Inflammatory liver diseases are a major cause of morbidity and mortality worldwide; however, underlying mechanisms are incompletely understood. Here we show that deleting the focal adhesion protein Kindlin-2 expression in hepatocytes using the *Alb-Cre* transgenic mice causes a severe inflammation, resulting in premature death. Kindlin-2 loss accelerates hepatocyte apoptosis with subsequent compensatory cell proliferation and accumulation of the collagenous extracellular matrix, leading to massive liver fibrosis and dysfunction. Mechanistically, Kindlin-2 loss abnormally activates the tumor necrosis factor (TNF) pathway. Blocking activation of the TNF signaling pathway by deleting TNF receptor or deletion of Caspase 8 expression in hepatocytes essentially restores liver function and prevents premature death caused by Kindlin-2 loss. Finally, of translational significance, adeno-associated virus mediated overexpression of Kindlin-2 in hepatocytes attenuates the D-galactosamine and lipopolysaccharide-induced liver injury and death in mice. Collectively, we establish that Kindlin-2 acts as a novel intrinsic inhibitor of the TNF pathway to maintain liver homeostasis and may define a useful therapeutic target for liver diseases.

## Editor's evaluation

Gao et al. developed various genetic permutations of mouse models of kindlin-2 deficiency in the hepatocytes to decipher its role. Hepatocyte-specific loss of kindlin-2 resulted in severe inflammatory liver injury, accelerated fibrosis/portal hypertension, and massive hepatocyte cell death by apoptosis. These effects are reversed by ablation of TNF signally or by caspase 8 deletion. AAV-mediated replacement of kindlin-2 protects the mice from chemically induced acute liver injury. Together the findings are novel with significant translational potential.

## Introduction

The liver is a multifunctional organ that plays critical roles in regulation of metabolism and detoxification and maintenance of the whole-body homeostasis (*Jones et al., 2018*). Liver comprised parenchymal cells and non-parenchymal cells. The former includes hepatocytes and endothelial cells; the latter includes hepatic stellate cells (HSC) and Kupffer cells. Abnormal activation of the TNF signaling stimulates inflammation and apoptosis, which are intertwined during the development and progression of liver diseases (*Guicciardi and Gores, 2005*; *Tacke and Zimmermann, 2014*; *Garcia-Martinez et al., 2016*; *Schuppan et al., 2018*). TNFα exerts its actions through binding and activating two different types of receptors, TNF receptor 1 (TNFR1) and TNFR2 (*Wajant and Siegmund, 2019*). It is known that NF-κB, a major TNFR downstream effector, plays a pivotal role in promoting inflammatory liver diseases, such as viral and alcoholic hepatitis and fulminant hepatic failure (*Ding and Yin, 2004*). Thus, it is critical to keep the TNF/NF-κB signaling activity under strict check to maintain normal organogenesis and function. However, key signal(s) that restrict abnormal TNF/TNFR activation in liver under physiological conditions and if and how alterations of these signals contribute to development and progression of inflammatory liver diseases are incompletely understood.

Kindlin proteins are key components of the focal adhesion (FA) assembly and contain the FERM domain, which is responsible for binding to and activating integrins to regulate the cell-extracellular matrix (ECM) adhesion, migration, and signaling (*Ma et al., 2008*; *Montanez et al., 2008*; *Hirbawi et al., 2017*; *Sun et al., 2019*; *Kadry and Calderwood, 2020*; *Michael and Parsons, 2020*). Mammals have three Kindlin proteins, that is, Kindlin-1, Kindlin-2, and Kindlin-3, which are encoded by *Fermt1*, *Fermt2*, and *Fermt3* genes, respectively. Kindlin-1 and Kindlin-3 are expressed primarily in epithelial and hematopoietic cells, respectively, while Kindlin-2 is widely expressed. Up-regulation of Kindlin-2 expression is implicated in a number of pathological processes, such as tumor formation, progression, and metastasis (*Zhan and Zhang, 2018*; *Sossey-Alaoui et al., 2019*). Global deletion of Kindlin-2 expression causes early embryonic lethality at E7.5 in mice (*Montanez et al., 2008*). Furthermore, Kindlin-2 and other FA-related proteins are widely involved in the control of the development and function of skeleton, kidney, heart, and other organs and tissues (*Wu et al., 2015*; *Zhang et al., 2016*; *Sun et al., 2017*; *Gao et al., 2019*; *Wang et al., 2019b*; *Zhang et al., 2019*; *Cao et al., 2020*; *Fu et al., 2020*; *He et al., 2020*; *Lei et al., 2020*; *Wegermann et al., 2020*; *Zhu et al., 2020*; *Eysert et al., 2021*; *Qin et al., 2021*; *Wang et al., 2021*; *Chen et al., 2022*; *Wu et al., 2022*) through both integrin-dependent and integrin-independent mechanisms. We have recently demonstrated that Kindlin-2 haploinsufficiency protects against non-alcoholic fatty liver disease by modulating the transcription factor Foxo1 in hepatocytes (*Gao et al., 2022*). However, whether and how Kindlin-2 plays a role during liver development is unknown.

In this study, by utilizing biochemical and genetic mouse approaches, we establish that Kindlin-2 acts as a potent inhibitor of the TNF/NF-κB-Caspase 8 pathway in hepatocytes and plays an important role in maintaining normal liver development and function.

## Results

### Kindlin-2 deletion in hepatocytes causes acute liver injury and premature death in mice

To investigate whether Kindlin-2 plays a role in liver development, we deleted its expression by crossing the floxed Kindlin-2 mice (*Fermt2*<sup>fl/fl</sup>) with *Alb-Cre* transgenic mice. The cross-breeding gave rise to genotypes at the expected Mendelian ratio at birth. Results from quantitative real-time RT-PCR and western blotting analyses revealed that the levels of *Fermt2* mRNA and protein were decreased in livers, but not brain, heart, lung, kidney, and spleen, in *Fermt2*<sup>fl/fl</sup>; *Alb-Cre* (KO) mice, compared to those in control littermates (*Figure 1—figure supplement 1*). Shockingly, all KO mice (>30 mice) died at ages between 4 and 5 weeks (*Figure 1a*). At 4 weeks of age, both body and liver weights of KO mice were slightly, but significantly, decreased compared to those of control littermates (*Figure 1b and c*). At this age, livers from KO mice became coarsened and granular (*Figure 1d*, top panels). Massive ascites was observed in all KO mice but not in control mice (*Figure 1d*, bottom panels). KO mice displayed a splenomegaly (*Figure 1e*) and severe cholestasis (*Figure 1f*). Compared to serum from control mice, serum from KO mice was more yellowish in color (*Figure 1g*) and displayed elevated serum levels of both direct and indirect bilirubin (*Figure 1h*), suggesting that the animals

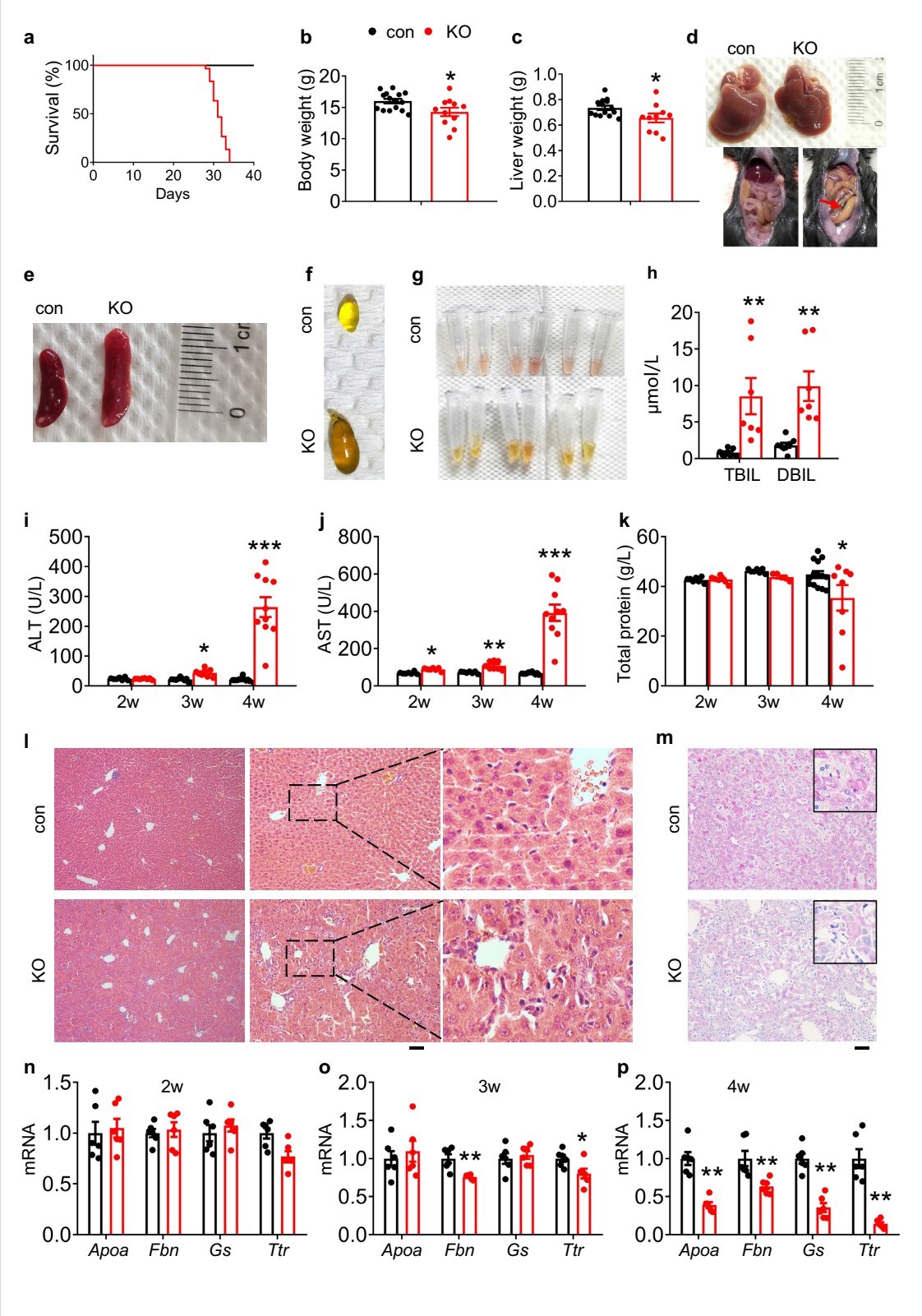

**Figure 1.** Kindlin-2 loss in hepatocytes causes liver injury and premature death in mice. (**a**) Survival curve of control and KO mice (N=30 mice/group). (**b**) Body weight of 4-week-old control and KO mice (N=15 for control mice, N=11 for KO mice). (**c**) Liver weight of 4-week-old control and KO mice (N=14 for control mice, N=10 mice for KO mice). (**d**) Livers and gross appearance of 4-week-old control and KO mice. Red arrow indicates massive ascites in KO. (**e**) Spleens from 4-week-old control and KO mice. (**f**) Gallbladder from 4-week-old control and KO mice. (**g**) Serum appearance from

*Figure 1 continued on next page*

*Figure 1 continued*

4-week-old control and KO mice. (**h**) Serum total bilirubin (TBIL) and direct bilirubin (DBIL) levels. (N=8 for control mice, N=7 mice for KO mice). (**i, j**) Serum aminotransferase (alanine transaminase [ALT] and aspartate transaminase [AST]) activity in 2-, 3-, and 4-week-old control and KO mice. (**k**) Serum total protein levels in 2-, 3-, and 4-week-old control and KO mice. (**l**) Hematoxylin and eosin staining of liver sections of 4-week-old control and KO mice. Scale bars,100 μm. (**m**) Periodic acid-Schiff staining for liver glycogen at 4 weeks of age. Scale bars,100 μm. (**n–p**) Quantitative real-time RT-PCR analysis of expression of liver genes in 2-, 3-, and 4-week-old control and KO mice (N=6 mice/group). The results are shown as means ± SEM. *p<0.05, **p<0.01, vs control.

The online version of this article includes the following source data and figure supplement(s) for figure 1:

**Source data 1.** Raw data related to *Figure 1*.

**Figure supplement 1.** Deletion of Kindlin-2 in hepatocyte resulted in systemic dysfunction.

**Figure supplement 1—source data 1.** Raw data related to *Figure 1—figure supplement 1*.

**Figure supplement 2.** Deletion of Kindlin-2 in hepatocyte resulted in osteoporosis.

**Figure supplement 2—source data 1.** Raw data related to *Figure 1—figure supplement 2*.

**Figure supplement 3.** Metabolic parameters were assessed on 4-week-old mice.

**Figure supplement 3—source data 1.** Raw data related to *Figure 1—figure supplement 3*.

suffered from cholestasis. KO mice displayed an osteopenic phenotype with significant reductions in both trabecular and cortical bone volume (*Figure 1—figure supplement 2*).

We further analyzed serum biochemistry in KO mice and age- and sex-matched littermates. The levels of both serum alanine transaminase (ALT) and aspartate transaminase (AST) activities were increased in KO mice relative to those in control mice, starting as early as 2 weeks of age with most dramatic increases at 4 weeks of age for both enzymes (*Figure 1i and j*). Furthermore, the levels of serum alkaline phosphatase (*Figure 1—figure supplement 3a*), γ-glutamyl transferase (*Figure 1—figure supplement 3b*) and total bile acid (*Figure 1—figure supplement 3c*) were increased in KO mice relative to those in control mice. The levels of serum protein were reduced in a time-dependent manner (*Figure 1k*). The level of serum low density lipoprotein was increased and that of serum high density lipoprotein was decreased in KO relative to control mice (*Figure 1—figure supplement 3d,e*). The levels of serum total cholesterol (TCH) and ammonia were increased in KO versus control mice (*Figure 1—figure supplement 3f,g*). Results from hematoxylin and eosin (H/E) staining of liver sections of 4-week-old control and KO mice revealed that Kindlin-2 loss caused massive inflammatory cell infiltration in KO liver, especially in areas surrounding the hepatic sinusoidal vessels (*Figure 1l*). Kindlin-2 loss reduced the glycogen accumulation, as determined by periodic acid-Schiff staining (*Figure 1m*) and the mRNA expression levels of liver-related genes, including those encoding *glutamine synthetase (Gs), apolipoprotein A-I (Apoa), fibrinogen b (Fbn)*, and *transthyretin (Ttr)* (*Figure 1n–p*).

Collectively, these results demonstrate that Kindlin-2 loss in hepatocytes causes a severe liver failure and damages in multiple, resulting in premature death.

## Kindlin-2 loss induces dramatic hepatocyte apoptosis and stimulates proliferation of both biliary cells and hepatic stellate cells in liver

A progressive increase in hepatocyte apoptosis was observed in KO mice, as demonstrated by TdT-mediated dUTP nick end labeling (TUNEL) staining (*Figure 2a and b* and *Figure 2—figure supplement 1a*). Consistently, immunohistochemical (IHC) staining showed that the immune reactivity of Caspase 3 protein was dramatically increased in KO relative to control livers (*Figure 2c*). The levels of both active Caspase 3 and p65 protein were increased in KO livers compared to those in control livers in a time-dependent manner (*Figure 2—figure supplement 1b and c*). Western blotting revealed that level of the pro-apoptotic Bax protein was time dependently increased in KO livers compared to that in control livers (*Figure 2d*). In contrast, expression of the anti-apoptotic Bcl-2 protein was largely down-regulated in a time-dependent manner by Kindlin-2 deficiency (*Figure 2d*). Western blotting showed that the levels of the proliferation cell nuclear antigen (Pcna) and cyclin D1 proteins were increased in KO livers compared with those in control livers at 3 and 4 weeks of ages but not at 2 weeks of age (*Figure 2e*). Results from IHC staining of control and KO liver sections using an anti-Ki67 antibody revealed an increase in cell proliferation in KO liver tissue (*Figure 2f and g*). Kindlin-2 loss apparently increased the number of cytokeratin 19 (CK19)-expressing cells, that is, the biliary/

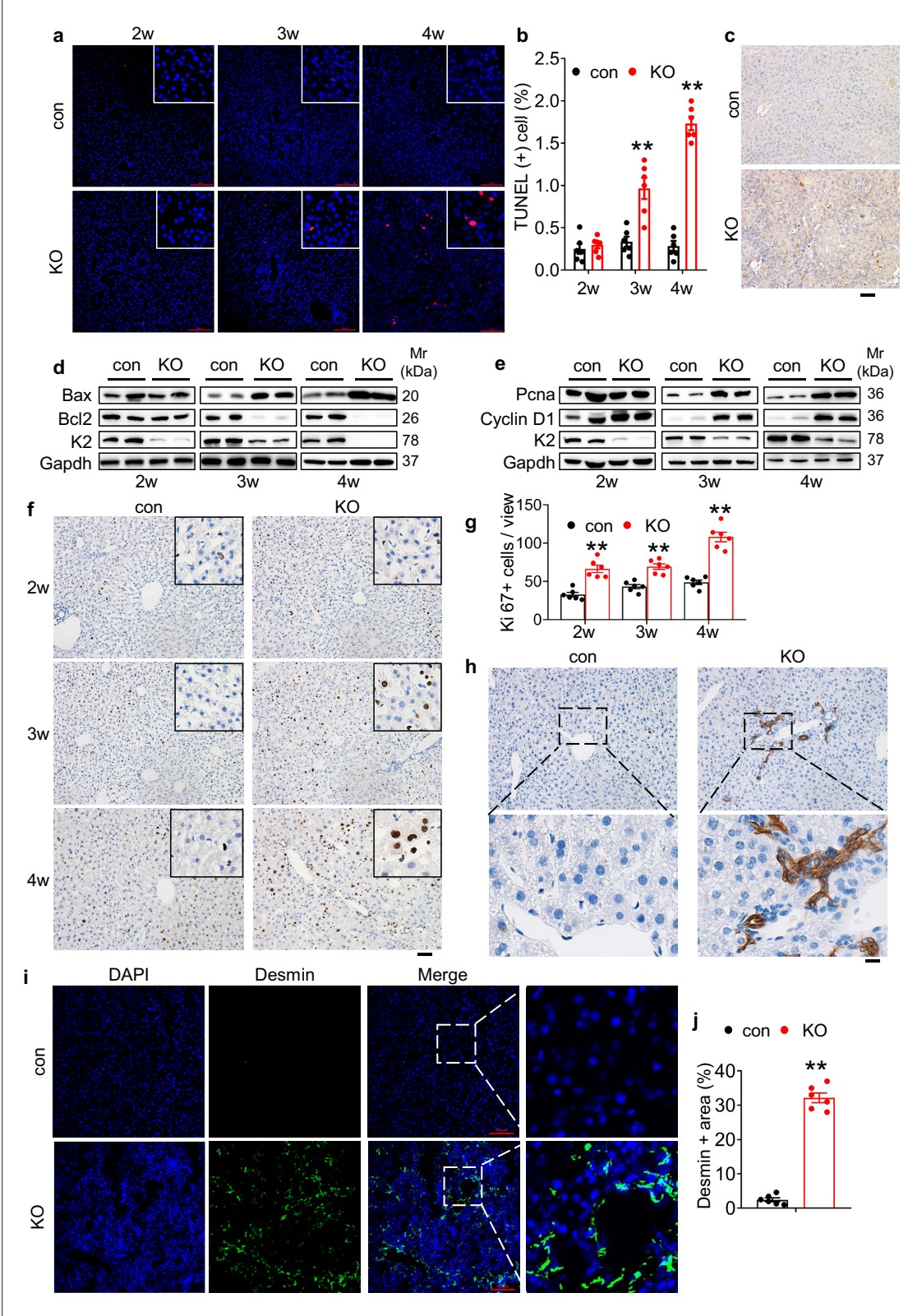

**Figure 2.** Kindlin-2 loss causes dramatic hepatocyte apoptosis followed by enhanced proliferation of both biliary cells and hepatic stellate cells.
(**a**) TUNEL staining. Scale bars,100 μm. (**b**) Quantification of TUNEL-positive cells from 6 different fields from 6 mice per group. (**c**) Immunohistochemistry
(IHC) staining for Caspase 3 was performed on paraffin liver sections of 4-week-old control and KO mice. Scale bars,100 μm. (**d, e**) Western blotting.
Liver tissue lysates from 2-, 3-, and 4-week-old control and KO mice were subjected to western blotting analysis for expression of the indicated proteins.

*Figure 2 continued on next page*

*Figure 2 continued*

Gapdh was used as a loading control. (**f, g**) Ki67 staining. Liver sections from of 2-, 3-, and 4-week-old control and KO mice were subjected to Ki67 staining. Scale bars,100 μm. Quantification of (**f**). (**h**) IHC staining. Liver sections from 4-week-old control and KO mice were subjected to IHC staining for CK19. Scale bars,100 μm. Representative images showing a dramatic increase in the number CK19-positive cells in KO liver. (**i, j**) Immunofluorescence (IF) staining. Scale bars,100 μm. Liver sections from 4-week-old control and KO mice were subjected to IF staining for Desmin, a marker for hepatic stellate cells. Representative images showing high expression of Desmin in KO liver. Quantification of (**i**) from six mice per group. The results are shown as means ± SEM. *p<0.05, **p<0.01, vs control.

The online version of this article includes the following source data and figure supplement(s) for figure 2:

**Source data 1.** Raw data related to *Figure 2*.

**Figure supplement 1.** Loss of Kindlin-2 resulted in increased apoptosis.

progenitor cells (*Figure 2h*). Immunofluorescence (IF) staining revealed that the number of Desmin-expressing cells, that is, HSC was dramatically increased in KO relative to that in control liver tissues (*Figure 2i and j*).

## Kindlin-2 loss stimulates collagen deposition and massive fibrosis in liver

A dramatic increase in collagen deposition was observed in livers of 4-week-old KO mice, as demonstrated by the Masson trichromatic and Sirius red staining (*Figure 3a–d*). Consistently, results from western blotting revealed that level of α-smooth muscle actin (α-Sma), a marker for fibrosis, was drastically increased in KO livers in a time-dependent manner (*Figure 3e*). IHC staining was also revealed that α-Sma was upregulated in KO mice (*Figure 3f*). In addition, the mRNA levels of fibrogenic genes, including those encoding *collagen α–1(I) chain (Col1a1)*, *transforming growth factor β1 (Tgfβ1)*, *TIMP metallopeptidase inhibitor 1 (Timp1)*, *actin α2 (Acta2)*, and *collagen type VI, α1 (Col6a1)*, were up-regulated in KO mice relative to those in control mice (*Figure 3g*).

## Kindlin-2 loss greatly activates the TNF/NF-κB signaling pathway in vitro and in vivo

We further performed RNA-seq analyses from 4-week-old control and KO liver tissues and identified 6746 genes that exhibited statistically significant changes and differential expression (3496 up-regulated genes and 3250 down-regulated genes with log2FC >1.5). Kyoto Encyclopedia of Genes and Genome (KEGG) analysis revealed that Kindlin-2 loss impacted multiple important pathways. Kindlin-2 loss activated TNF signaling pathway, apoptosis, and MAPK pathways in hepatocytes (*Figure 4a and b*). Consistent with result from the RNA-seq analysis, the serum level of tumor necrosis factor α (Tnfα) was dramatically elevated in 4-week-old KO mice compared to that in control littermates (*Figure 4c*). Furthermore, the mRNA levels of *Tnfα* were increased in a time-dependent manner in KO livers compared to those in control livers (*Figure 4d*). IHC staining of F4/80 revealed massive macrophage infiltration in KO but not control liver tissues (*Figure 4e*). In addition, the expression of CD19 (B cell marker) and ly6G (neutrophils marker) was increased in KO versus control livers (*Figure 4f and g*). The numbers of circulating neutrophils, myeloid cells (e.g. monocytes), and blood platelet counts were increased in KO versus control mice (*Figure 4h–j* and *Figure 4—figure supplement 1*).

To define the function of Kindlin-2 in hepatocyte in vitro, we knocked down its expression in Huh7 and HepG2 cells. The levels of *Fermt2* mRNA and Kindlin-2 protein were dramatically reduced by *Fermt2* shRNA (sh-K2) in both Huh7 (*Figure 4k and l*) and HepG2 cells (*Figure 4—figure supplement 2a,b*). The mRNA levels of *Tnfα* and *Birc3 (Baculoviral IAP repeat containing 3)* were up-regulated by Kindlin-2 knockdown in Huh7 (*Figure 4m*) and HepG2 cells (*Figure 4—figure supplement 2c*). ELISA showed a dramatic increase in TNF protein in supernatants of the sh-K2 Huh7 (*Figure 4n*) and HepG2 cultures (*Figure 4—figure supplement 2d*). Results from F-actin IF staining showed that Kindlin-2 loss impaired the FA formation and caused a reduced tension (*Figure 4o*). Moreover, deletion of Kindlin-2 in hepatocyte resulted in defects in cell spreading and attachment (*Figure 4p*).

Collectively, these results illustrate that ablation of Kindlin-2 in hepatocyte resulted in upregulation of TNF signaling pathway.

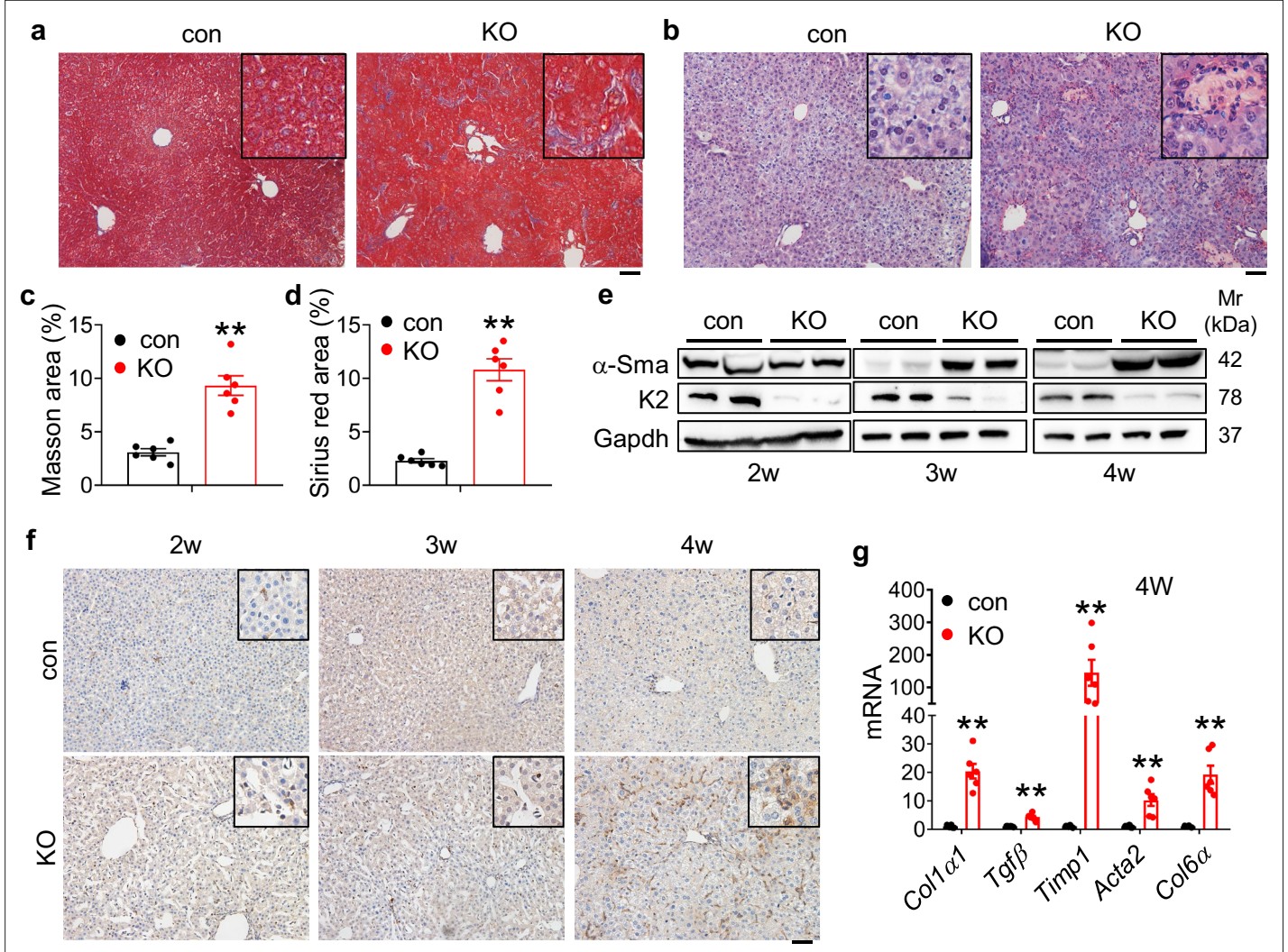

**Figure 3.** Kindlin-2 loss promotes collagen extracellular matrix deposition and fibrosis in liver. (**a**) Masson's trichrome staining of liver sections of 4-week-old control and KO mice. Scale bars,100 μm. (**b**) Sirius red staining. Fibrillar collagen deposition in liver sections of 4-week-old control and KO mice was determined by Sirius red staining. Scale bars,100 μm. (**c, d**) Quantification of (**a**) and (**b**) from six mice per group. (**e**) Western blotting. Liver tissue extracts from 2-, 3-, and 4-week-old control and KO mice were subjected to western blotting for expression of α-Sma and Kindlin-2. Gapdh was used as loading control. (**f**) Immunohistochemistry (IHC) staining. Liver sections from 2-, 3-, and 4-week-old control and KO mice were subjected to IHC staining using an antibody against α-Sma. Scale bars,100 μm. (**g**) Quantitative real-time RT-PCR (qRT-PCR) analysis. RNAs isolated from liver tissues of 4-week-old control and KO mice were subjected to qRT-PCR analysis for expression of fibrosis-related genes. (N=6 mice/group). The results are shown as means ± SEM. *p<0.05, **p<0.01, vs control.

The online version of this article includes the following source data for figure 3:

**Source data 1.** Raw data related to *Figure 3*.

## TNFR ablation reverses liver lesions and lethality caused by Kindlin-2 deficiency

To determine whether abnormal activation of the TNF signaling plays a key role in mediating the liver damage and death caused by Kindlin-2 deficiency, we globally deleted the expression of the TNF receptors in Kindlin-2 KO mice and determined its effects. We used a breeding strategy by crossing the *Tnfr*[-/-] mice, in which both *Tnfrsf1a* and *Tnfrsf1b* are globally deleted, with the *Alb-Cre; Fermt2*[fl/+] mice to generate the Tnfr-deficient Kindlin-2 KO mice (referred to as KO/*Tnfr*[-/-] hereafter) (*Figure 5—figure supplement 1*). *Tnfr*[-/-] mice were purchased from Shanghai Biomodel Organism Science & Technology Development Co., Ltd. The KO/*Tnfr*[-/-] mice were normal at birth with the expected Mendelian ratio. Strikingly, deletion of *Tnfr* genes completely prevented the premature death of the Kindlin-2

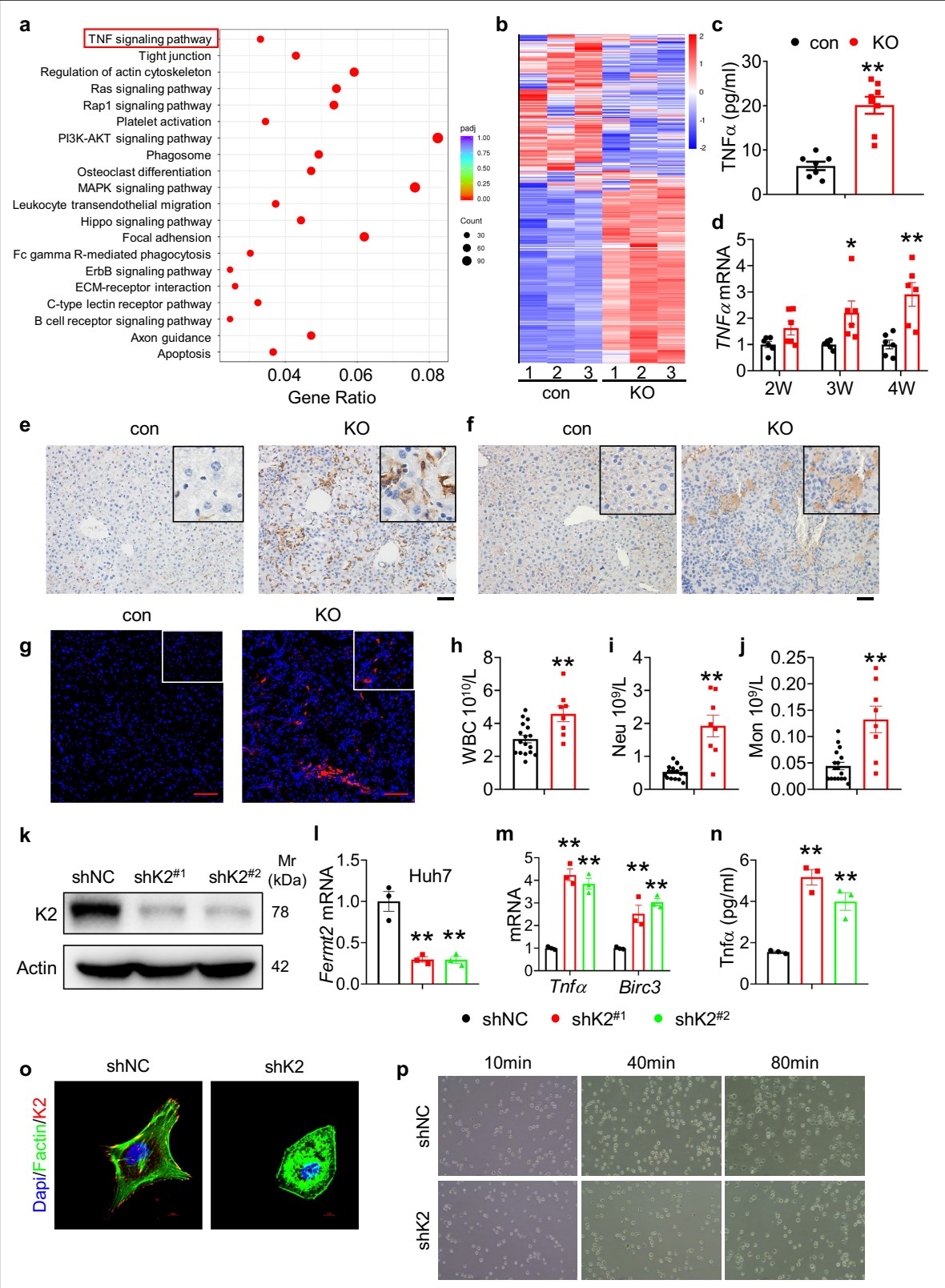

**Figure 4.** Kindlin-2 loss activates the TNF/NF-$\kappa$B signaling pathway. (**a**) Kyoto Encyclopedia of Genes and Genome analysis showing the up-regulated pathways in the 4-week-old KO mice versus control mice. (**b**) Heatmap represents genes with >1.5 fold upregulation or >1.5 fold downregulation in K2-deficient liver compared with control liver. (**c**) Serum from 4-week-old control and KO mice was subjected to ELISA for Tnfα protein (N=7–8 mice/group). (**d**) RNAs isolated from liver tissues of 2-, 3-, and 4-week-old control and KO mice were subjected to quantitative real-time RT-PCR (qRT-PCR) analyses

*Figure 4 continued on next page*

*Figure 4 continued*

for *Tnfα* genes (N=6 mice/group). (**e**) 4-week-old control and KO mouse liver sections were subjected to immunohistochemistry (IHC) staining using an anti-F4/80 antibody to determine macrophage infiltration. Scale bars,100 μm. (**f**) 4-week-old control and KO mouse liver sections were subjected to IHC staining using an anti-CD19 antibody to determine B cell infiltration. Scale bars,100 μm. (**g**) 4-week-old control and KO mouse liver sections were subjected to immunofluorescence (IF) staining using an anti-Ly6G antibody to determine neutrophils infiltration. Scale bars,100 μm. (**h–j**) Complete blood count was assessed on 4-week-old mice. WBC: white blood cell, Neu: neutrophil, Mon: monocyte. N=17 for control mice and N=8 for KO mice. (**k, l**) Kindlin-2 knockdown. qRT-PCR analyses and western blotting were performed to detect Kindlin-2 expression in Huh7 cells treated with lentiviruses-expressed control shRNA (shNC) and two different *Fermt2* shRNAs (shK2[#1], shK2[#2]). (**m**) qRT-PCR analysis of *Tnfα* and *Bircl3* mRNAs in shNC- and shK2-treated Huh7 cells. (**n**) ELISA. The levels of Tnfα protein in media of shNC- and shK2-treated Huh7 cultures were assayed using an ELISA kit. (**o**) (IF) staining for DAPI, F-actin, and Kindlin-2 in shNC and shK2 cells. (**p**) Representative images of shNC and shK2 cells spreading and attachment on glass surface at the indicated times after seeding. Each sample was tested at least in triplicate independent cell preparations.The results are shown as means ± SEM. *p<0.05, **p<0.01, vs control.

The online version of this article includes the following source data and figure supplement(s) for figure 4:

**Source data 1.** Raw data related to *Figure 4*.

**Figure supplement 1.** Complete blood count was assessed on 4-week-old mice.

**Figure supplement 1—source data 1.** Raw data related to *Figure 4—figure supplement 1*.

**Figure supplement 2.** Kindlin-2 loss in HepG2 cells resulted in upregulation of TNF signaling pathway.

**Figure supplement 2—source data 1.** Raw data related to *Figure 4—figure supplement 2*.

KO mice. In contrast to the fact that all Kindlin-2 KO mice died between 4 and 5 weeks of ages, no KO/*Tnfr*[-/-] mice (>30 mice) died during this period of time and beyond (*Figure 5a*). The levels of serum ALT and AST were significantly lower in KO/*Tnfr*[-/-] mice than in Kindlin-2 KO mice (*Figure 5b and c*). As shown in *Figure 5d*, deletion of *Tnfr* genes rescued the liver damage in Kindlin-2 KO mice. The increases in collagen deposition, macrophage infiltration, cell proliferation, fibrosis, and apoptosis caused by Kindlin-2 deficiency were markedly reversed by *Tnfr* genes deletion (*Figure 5e–n*).

## Caspase 8 deletion in hepatocytes attenuates liver lesions and prevents lethality of Kindlin-2 KO mice

It is known that abnormal inflammation causes cell death, and that Caspase 8 is a major initiator in the death receptor-mediated apoptosis. We therefore next determined whether Kindlin-2 loss causes hepatocyte death and thereby liver lesions and lethality by activating the Caspase 8-dependent apoptotic pathway. To this end, we generated mice lacking both Kindlin-2 and Caspase 8 in hepatocytes (referred to as KO/*Casp8*[-/-] hereafter). Shockingly, deletion of Caspase 8 in hepatocytes essentially prevented premature death of Kindlin-2 KO mice (*Figure 6a*). The levels of serum ALT and AST were significantly lower in KO/*Casp8*[-/-] mice than in Kindlin-2 KO mice (*Figure 6b and c*). Deletion of Caspase 8 restored the liver damage in Kindlin-2 KO mice (*Figure 6d*). The increases in macrophage infiltration, cell proliferation, fibrosis, and apoptosis caused by Kindlin-2 deficiency were markedly attenuated by Caspase 8 deletion (*Figure 6e–l*).

## AAV8-mediated overexpression of Kindlin-2 in hepatocytes attenuates the GalN/LPS-induced liver injury and death

We finally investigated the effect of Kindlin-2 on the liver injury induced by D-galactosamine (GalN)/lipopolysaccharide (LPS). Immunoblotting analysis detected a dramatic decrease in Kindlin-2 accumulation in the liver tissues of C57BL/6 after administration with GalN/LPS (*Figure 7a and b*). Next, 8-week-old C57BL/6 mice were first injected via tail vein with adeno-associated virus 8 (AAV8) ($2 \times 10^{11}$ particles/mouse) expressing Kindlin-2 (AAV8-K2) or (Green fluorescent protein) GFP (AAV8-GFP). After 21 days, mice were treated with GalN/LPS for 5 hr or a longer period of time (for survival curve experiment) (*Figure 7c*). The level of Kindlin-2 protein was markedly increased in livers of mice injected with AAV8-K2 (*Figure 7d and e*). As expected, GalN/LPS resulted in acute death of all mice around 14–15 hr of the treatment. Kindlin-2 overexpression (OE) significantly extended the life span of mice compared with control mice injected with AAV8-GFP (*Figure 7f*). Kindlin-2 OE improved the gross liver appearance in GalN/LPS-treated mice (*Figure 7g*). Kindlin-2 OE significantly decreased the levels of serum ALT and AST in GalN/LPS-treated mice (*Figure 7h and i*), but no difference in untreated Kindlin-2 OE and control mice (*Figure 7—figure supplement 1*). Results from H/E staining

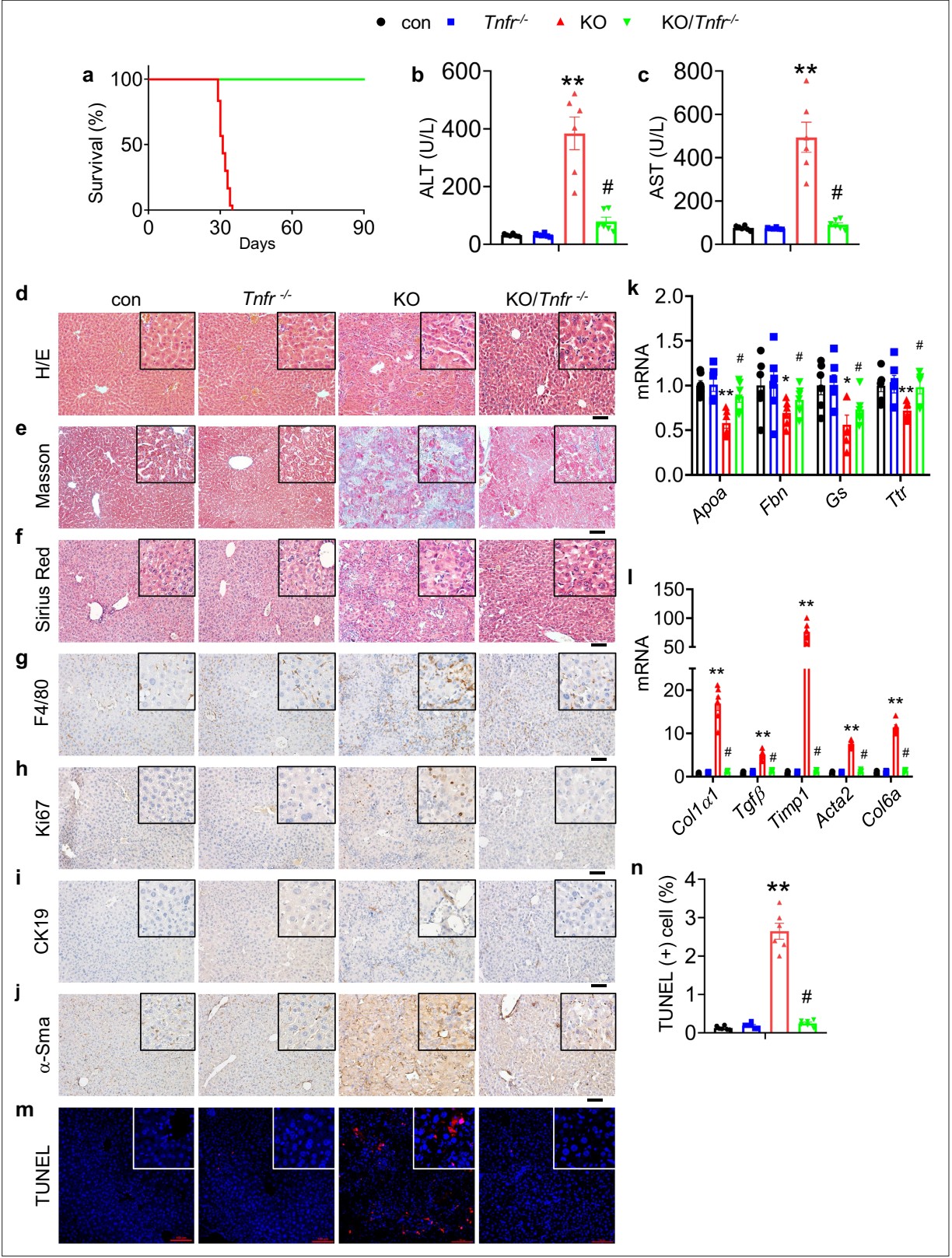

**Figure 5.** Global *Tnfr* genes deletion rescues the liver injury and lethality induced by Kindlin-2 loss. (**a**) Survival curve of KO and KO/*Tnfr*⁻/⁻ mice. N=30 mice in KO group, N=33 for KO/*Tnfr*⁻/⁻ group. (**b, c**) Serum alanine transaminase (ALT) and aspartate transaminase (AST) levels in control, KO, *Tnfr*⁻/⁻, and KO/*Tnfr*⁻/⁻ mice. N=6 mice/group. (**d-f**) Liver histology. Liver sections from mice of the indicated genotypes were subjected to hematoxylin and eosin (H/E) staining (**d**), Masson's trichrome staining (**e**) and Sirius Red staining (**f**). Scale bars,100 μm. (**g–j**) Immunohistochemistry staining of

*Figure 5 continued on next page*

*Figure 5 continued*

liver sections from mice of the indicated genotypes for expression of F4/80 (**g**), Ki67 (**h**), CK19 (**i**), and α-Sma (**j**). Scale bars,100 μm. (**k, l**) Quantitative real-time RT-PCR (qRT-PCR) analysis. RNAs isolated from liver tissues of the indicated genotypes were subjected to qRT-PCR analysis for expression of the indicated genes. N=6 mice/group. (**m, n**) TUNEL staining of the indicated genotypes and quantification. N=6 mice/group. Scale bars,100 μm. The results are shown as means ± SEM. *p<0.05, **p<0.01, vs control. #p<0.05, vs KO.

The online version of this article includes the following source data and figure supplement(s) for figure 5:

**Source data 1.** Raw data related to *Figure 5*.

**Figure supplement 1.** Breeding strategy to generate the *Alb-Cre; Fermt2*$^{fl/+}$*; Tnfrsf*$^{-/-}$ mice.

revealed that histological damages in liver caused by GalN/LPS were markedly ameliorated by Kindlin-2 OE (*Figure 7j*). Furthermore, Kindlin-2 OE decreased GalN/LPS-induced hepatocyte apoptosis, as measured by IHC staining of Caspase-3 and TUNEL staining of liver sections (*Figure 7k–m*).

## Discussion

In this study, we demonstrate a novel role of the key FA protein Kindlin-2 in regulation of liver development and function. Kindlin-2 loss in hepatocytes causes an acute liver failure and premature death in mice. We demonstrate that Kindlin-2 loss dramatically activates the TNF/NF-κB signaling pathway, massive inflammation, and hepatocyte death with subsequent stimulation of proliferation of both biliary cells and HSCs, leading to ECM accumulation and fibrosis in liver. We further demonstrate that liver damages and lethality caused by Kindlin-2 deficiency are largely attenuated or completely rescued by either global ablation of TNFR or hepatocyte-selective deletion of Caspase 8 expression. We finally demonstrate that adeno-associated virus mediated OE of Kindlin-2 in hepatocytes attenuates the GalN/LPS-induced liver injury and death in mice.

In this study, we demonstrate that Kindlin-2 expression in hepatocytes is essential for liver development and function. Kindlin-2 deletion in hepatocytes using the *Alb-Cre* transgenic mice causes acute liver failure and lethality of 100% penetration in mice at ages between 4 and 5 weeks. Our results suggest that abnormal activation of the TNF/NF-κB signaling pathway plays a major role in mediation of the liver lesions and lethality caused by Kindlin-2 deficiency. This notion is supported by the following lines of in vitro and in vivo evidence. First, results from our RNA-seq analyses from 4-week-old control and KO liver tissues reveal that TNF signaling pathway is activated in KO liver. Second, the levels of *Tnfα* mRNA in liver and serum level of TNFα protein were dramatically elevated in 4-week-old KO versus control mice. Consistently, massive macrophage infiltration is observed in KO but not control liver tissues, suggesting increased inflammation. Third, in vitro studies show that shRNA knockdown of *Fermt2* expression increases the mRNA levels of *Tnfα* and *Birc3* in Huh7 and HepG2 cells. Fourth, the increases in serum ALT and AST, hepatocyte apoptosis, ECM deposition, inflammatory infiltration, proliferation of both biliary cells and HSCs, and liver fibrosis caused by Kindlin-2 deficiency were largely reversed by *Tnfr* genes deletion. Most importantly, *Tnfr* genes deletion completely rescues the lethality caused by Kindlin-2 deficiency. These findings also indicate the necessity to keep the TNF/NF-κB signaling pathway in hepatocytes under precise control in order to maintain normal liver development and function as well as the whole-body metabolic homeostasis. We demonstrate that Kindlin-2 plays a pivotal role in this regard.

Kindlin-2 loss induces massive hepatocyte apoptosis. The expression of the pro-apoptotic Bax protein is up-regulated and that of the anti-apoptotic Bcl2 protein is down-regulated in mutant liver. A number of apoptotic hepatocytes are observed throughout the mutant liver. Strikingly, genetic ablation of *Caspase 8* expression in hepatocytes essentially rescues the liver lesions and lethality caused by Kindlin-2 loss. These results suggest that the Caspase 8-dependent death receptor-mediated apoptotic pathway plays a critical role in mediating the liver damages and lethality caused by Kindlin-2 loss. Abnormal inflammation induced by Kindlin-2 loss should largely contribute to hepatocyte apoptosis in the mutant mice.

Notably, Kindlin-2 loss greatly promotes proliferation of both biliary cells and HSCs and stimulates accumulation and deposition of excessive collagenous ECM, which leads to liver fibrosis in the mutant mice.

It should be noted that KO mice display a splenomegaly and severe cholestasis. These results along with the severe fibrosis strongly suggest the presence of portal hypertension in the mutant mice. It

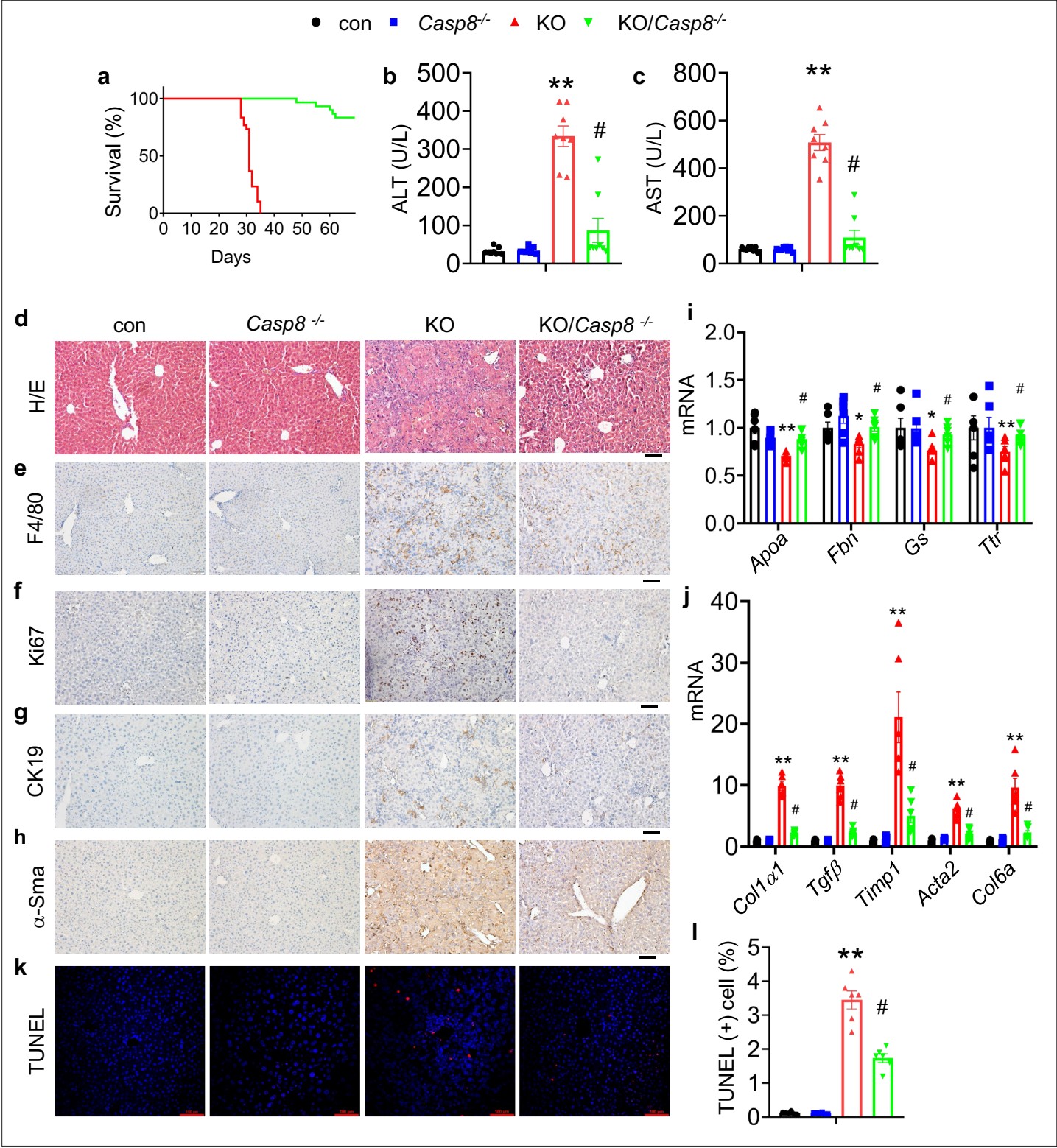

**Figure 6.** Caspase 8 deletion in hepatocytes rescues the liver lesions and lethality caused by Kindlin-2 deficiency. (**a**) Survival curve of KO and KO/*Casp8⁻/⁻* mice. N=30 mice/group. (**b, c**) Serum alanine transaminase (ALT) and aspartate transaminase (AST) levels in control, KO, *Casp8⁻/⁻*, and KO/*Casp8⁻/⁻* mice. N=8 mice/group. (**d**) Liver histology. Liver sections from mice of the indicated genotypes were subjected to hematoxylin and eosin (H/E) staining, Scale bars,100 μm. (**e–h**) Immunohistochemistry staining of liver sections from mice of the indicated genotypes for expression of F4/80 (**e**), Ki67 (**f**), CK19 (**g**), and α-Sma (**h**). Scale bars,100 μm. (**i, j**) Quantitative real-time RT-PCR (qRT-PCR) analysis. RNAs isolated from liver tissues of the indicated genotypes were subjected to qRT-PCR analysis for expression of the indicated genes. N=6 mice/group. (**k, l**) TUNEL staining of the indicated

*Figure 6 continued on next page*

*Figure 6 continued*

genotypes and quantification. N=6 mice/group. Scale bars,100 μm. The results are shown as means ± SEM. *p<0.05, **p<0.01, vs control. #p<0.05, vs KO.

The online version of this article includes the following source data for figure 6:

**Source data 1.** Raw data related to *Figure 6*.

should also be noted that Kindlin-2 loss causes a significant reduction in bone mass. This could be in part due to the dyslipidemia induced by liver dysfunction, which is known to induce hepatic osteodystrophy (*Zaidi et al., 2022*).

Of translational significance, AAV8-mediated expression of Kindlin-2 in liver significantly blocks the D-GalN/LPS-induced liver inflammation and death in mice. In the future, it would be important to investigate whether Kindlin-2 loss plays a role in the pathogenesis of human inflammatory liver diseases.

In summary, we establish that the FA protein Kindlin-2 is a potent intrinsic inhibitor of the TNF/NF-κB-Caspase 8 inflammatory pathway and plays an important role in the maintenance of normal liver development and function. We may define a novel therapeutic target for inflammatory liver diseases.

## Materials and methods

### Animal study

Generation of *Fermt2*$^{fl/fl}$ mice was previously described (*Gao et al., 2019*). To delete Kindlin-2 expression in hepatocyte, we first bred the *Fermt2*$^{fl/fl}$ mice with the *Alb-Cre* transgenic mice (*Wang et al., 2019a*), which were kindly provided by Dr. Yan Li of Southern University of Science and Technology, and obtained the *Fermt2*$^{fl/+}$; *Alb-Cre* mice. Further intercrossing of the *Fermt2*$^{fl/+}$; *Alb-Cre* mice with *Fermt2*$^{fl/fl}$ mice generated the *Fermt2*$^{fl/fl}$; *Alb-Cre* mice, the hepatocyte conditional Kindlin-2 knockout mice (referred to as KO). The Cre-negative floxed Kindlin-2 mice (*Fermt2*$^{fl/fl}$) were used as controls in this study. *Tnfrsf1a*$^{-/-}$ and *Tnfrsf1b*$^{-/-}$ mice were obtained from the Shanghai Model Organism. *Caspase 8*$^{fl/fl}$ (*Casp8*) mice were used as indicated (*Gao et al., 2021*). All mice used in this study have been crossed with normal C57BL/6 mice for more than 10 generations. Mice were maintained in 12 hr light/dark cycles, with unrestricted access to food and water. All animal experiments were approved and conducted in the specific pathogen free Experimental Animal Center of Southern University of Science and Technology (Approval number: 20200074).

### Biochemical measurements

Blood and tissues were collected from mice under anesthesia with isoflurane. Blood collected was allowed to clot for 2 hr at room temperature and then centrifuged to collect serum. TCH, AST, ALT, and albumin were measured with commercial kits (Shensuoyoufu, Shanghai, China).

### Histological analyses

Tissues were fixed in 4% PFA and then embedded in paraffin. Serial 5 μm paraffin sections were used for H/E staining using our standard protocols. Masson trichrome staining and Sirius Red staining were performed as described previously (*Ni et al., 2014*). Immunohistochemistry was performed using paraffin sections according to a protocol previously described (*Cao et al., 2020*). In brief, samples were deparaffinized and rehydrated, followed by antigen retrieval in 10 mM sodium citrate. Blocking and staining were performed in antibody diluent with background-reducing components (DAKO). Samples were incubated with primary antibodies as listed in *Supplementary file 1*, followed by appropriate secondary antibodies. Images were obtained using a light microscope equipped with a digital camera.

### Immunofluorescence staining

IF staining was performed as previously described (*Cao et al., 2020*). Briefly, liver frozen sections (10 μm thickness) were prepared using a Leica cryostat, fixed in 4% paraformaldehyde for 30 min, blocked for 2 hr with 5% normal goat serum supplemented with 1% BSA, and incubated with the

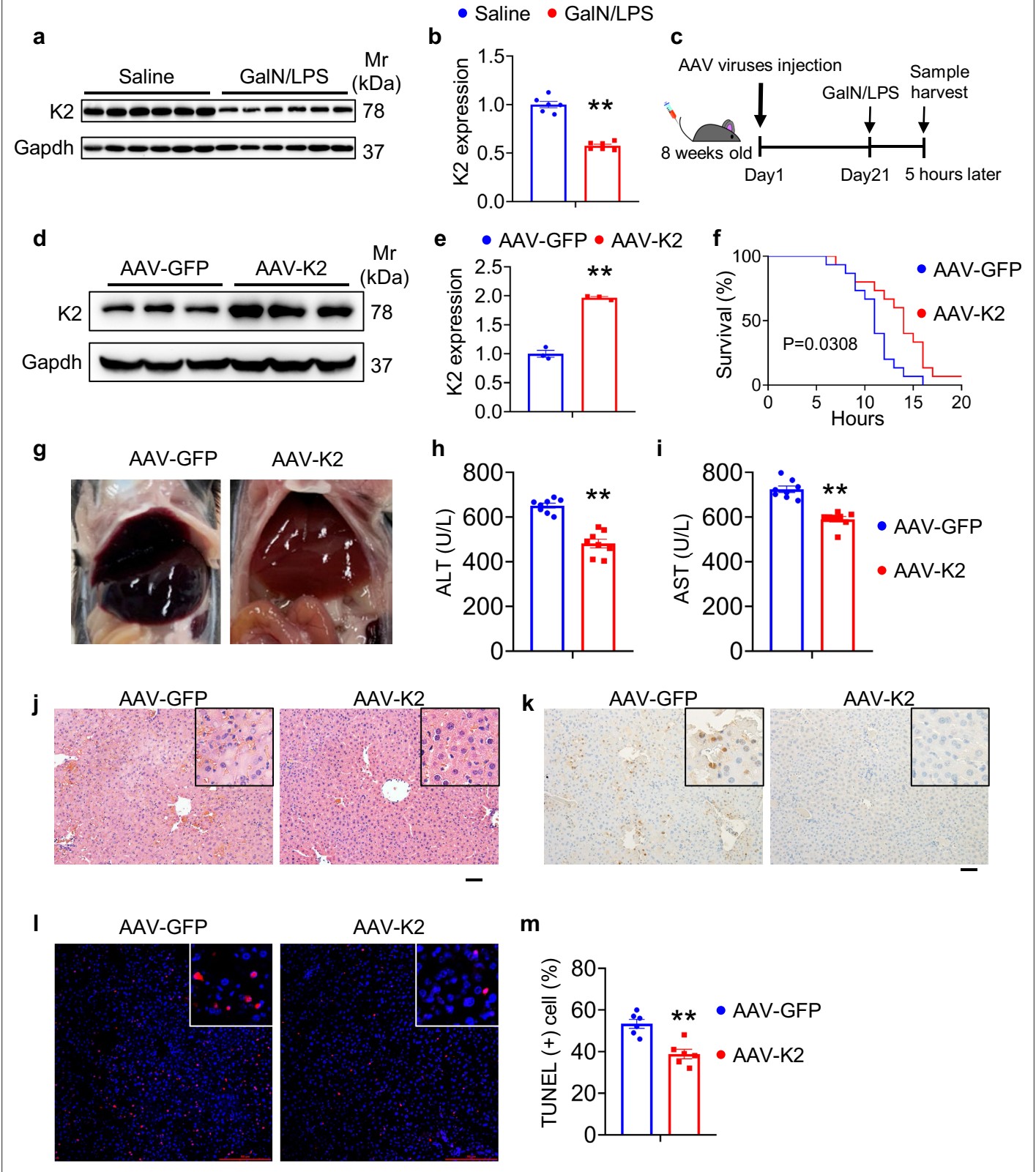

**Figure 7.** Overexpression of Kindlin-2 ameliorates D-galactosamine/lipopolysaccharide (D-GalN/LPS)-induced liver injury and death. (**a, b**) Eight-week-old C57BL/6 mice were intraperitoneally injected with D-GalN/LPS (GalN, 700 mg/kg body weight, LPS, 3 mg/kg body weight) or PBS as controls for 5 hr. Liver tissues were collected and subjected to western blotting for expression of Kindlin-2. Quantitative data (**a**). N=6 mice/group. (**c**) Experimental design. (**d, e**) Western blotting. Eight-week-old C57BL/6 mice were first injected via tail vein with adeno-associated virus 8 (AAV8) (2×10¹¹ particles/

*Figure 7 continued on next page*

*Figure 7 continued*

mouse) expressing Kindlin-2 (AAV8-K2) or GFP (AAV8-GFP). After 21 days, mice were then treated with D-GalN/LPS for 5 hr. Protein extracts were then prepared from livers and subjected to western blotting for expression of Kindlin-2. Quantitative data (**d**). Gapdh was used as a loading control. N=3 mice per group. (**f**) Survival curve. Mice were treated as in (**c**), followed by observation for death. N=15 mice per group. (**g**) Gross liver appearance. Mice were treated as in (**c**). (**h, i**) ELISA assays. Serum ALT (**h**) and AST (**i**). Mice were treated as in (**c**). N=8 mice per group. (**j**) Hematoxylin and eosin staining in liver sections. Mice were treated as in (**c**). (**k**) Immunohistochemistry staining of Caspase-3 in liver sections. Mice were treated as in (**c**). (**l, m**) TUNEL staining of liver sections. Quantitation data (**m**). Mice were treated as in (**c**). N=6 mice per group. The results are shown as means ± SEM. **p<0.01, vs AAV-GFP.

The online version of this article includes the following source data and figure supplement(s) for figure 7:

**Source data 1.** Raw data related to *Figure 7*.

**Figure supplement 1.** Serum alanine transaminase (ALT) and aspartate transaminase (AST).

**Figure supplement 1—source data 1.** Raw data related to *Figure 7—figure supplement 1*.

indicated antibodies at 4°C overnight. The sections were incubated with appropriate fluorescence-labeled secondary antibodies and analyzed by IF microscopy.

## Cell culture in vitro

HepG2 (cat# SCSP-510) and Huh7 (cat# TCHu182) cells were purchased from the Cell Bank of the Chinese Academy of Sciences and cultured in DMEM supplemented with 10% fetal bovine serum, 1% penicillin, and streptomycin in a 5% $CO_2$ incubator at 37°C. All cell lines tested negative for myco-plasma contamination.

## Western blot analysis

For western blotting, total protein samples were extracted from tissues or cells, and 30 μg of protein samples were separated on a 10% sodium dodecyl sulfate polyacrylamide gel electrophoresis gel and transferred to a polyvinylidene fluoride membrane. Protein expression was visualized by incubating primary antibodies overnight at 4°C, followed by the corresponding secondary antibodies, and developed using the enhanced chemiluminescence system (Bio-Rad, #1705040) (*Gao et al., 2012*; *Gao et al., 2021*). Antibodies information is described in *Supplementary file 1a*.

## RNA-seq data and GO/KEGG enrichment analyses

Total RNAs were extracted from livers of 4-week-old control and KO mice. The sequencing library was determined by an Agilent 2100 Bioanalyzer using the Agilent DNA 1000 Chip Kit (Agilent, #5067–1504). Differentially expressed genes (DEGs) were selected with p-value <0.05, log2FC >1 or < −1, and FPKM >5 in at least one condition. Volcano plot showing dysregulated genes between control and KO livers was generated using R programming. GO and KEGG enrichment analyses were performed on the DEGs using TBtools.

## Quantitative real-time RT-PCR analysis

Total RNA from tissues and cells was extracted with Trizol reagent (Invitrogen, #10296010) as described (*Gao et al., 2017*). After total RNA isolation and cDNA synthesis, PCR amplification was performed with the SYBR Green PCR Master Mix (Bio-Rad, #1725200). The mRNA expression levels of the target genes were normalized to expression of the *glyceraldehyde 3-phosphate dehydrogenase* (*Gapdh*). Each sample was tested at least in triplicate and repeated using three independent cell preparations. Primer sequences are listed in the *Supplementary file 1b* and *Supplementary file 1c*.

## TUNEL assay

Liver frozen sections were fixed with 4% paraformaldehyde and subjected to TUNEL staining using an In Situ Cell Death Detection Kit (Beyotime Biotechnology, #C1089), following manufacturer's procedure.

## Enzyme-linked immunosorbent assay (ELISA)

After collection of serum from mice and supernatants from cultured hepatocytes, TNFα (R&D System, #MTA00B and DTA00D) and Il1β (R&D System, #MLB00C) levels were measured by ELISA kit according to manufacturer's instruction.

## Statistical analysis

The sample size in mouse experiments of this study was determined based on our previous experience. Mice were randomly grouped in experiments in this study. IF, IHC, and histology were performed and analyzed in a double-blinded way. The two-tailed unpaired Student's $t$ test (two groups) and one-way ANOVA (multiple groups), followed by Tukey's post-hoc test, were used for statistical analyses using Prism GraphPad. The results are presented as means ± SEM (standard error of mean). $p < 0.05$ was considered statistically significant.

## Acknowledgements

The authors acknowledge the assistance of Core Research Facilities of SUSTech. This work was in part supported by the National Key Research and Development Program of China Grants (2019YFA0906004), the National Natural Science Foundation of China Grants (82230081, 82250710175, 82172375, 81991513 and 81870532), the Shenzhen Municipal Science and Technology Innovation Council Grants (JCYJ20220818100617036, JCYJ20180302174246105 and ZDSYS20140509142721429) and the Guangdong Provincial Science and Technology Innovation Council Grant (2017B030301018).

## Additional information

### Competing interests

Di Chen: Reviewing editor, *eLife*. The other authors declare that no competing interests exist.

### Funding

| Funder | Grant reference number | Author |
|---|---|---|
| National Key Research and Development Program of China | 2019YFA0906004 | Guozhi Xiao |
| National Natural Science Foundation of China | 82230081 | Guozhi Xiao |
| Shenzhen Municipal Science and Technology Innovation Council | JCYJ20180302174246105 | Huanqing Gao |
| Shenzhen Municipal Science and Technology Innovation Council | JCYJ20220818100617036 | Guozhi Xiao |
| Guangdong Provincial Science and Technology Innovation Council Grant | 2017B030301018 | Guozhi Xiao |
| Shenzhen Municipal Science and Technology Innovation Council | ZDSYS20140509142721429 | Guozhi Xiao |
| National Natural Science Foundation of China | 82250710175 | Guozhi Xiao |
| National Natural Science Foundation of China | 82172375 | Guozhi Xiao |
| National Natural Science Foundation of China | 81991513 | Guozhi Xiao |

| Funder | Grant reference number | Author |
| --- | --- | --- |
| National Natural Science Foundation of China | 81870532 | Guozhi Xiao |

The funders had no role in study design, data collection and interpretation, or the decision to submit the work for publication.

## Author contributions
Huanqing Gao, Conceptualization, Formal analysis, Supervision, Investigation, Writing - original draft; Yiming Zhong, Formal analysis, Investigation, Methodology, Writing - original draft; Liang Zhou, Formal analysis, Investigation, Methodology; Sixiong Lin, Xiaoting Hou, Zhen Ding, Investigation, Methodology; Yan Li, Resources; Qing Yao, Validation; Huiling Cao, Supervision; Xuenong Zou, Validation, Writing - review and editing; Di Chen, Xiaochun Bai, Supervision, Writing - review and editing; Guozhi Xiao, Conceptualization, Supervision, Funding acquisition, Writing - review and editing

## Author ORCIDs
Huanqing Gao ⓘ http://orcid.org/0000-0002-8567-3583
Yiming Zhong ⓘ http://orcid.org/0000-0002-7243-6503
Sixiong Lin ⓘ http://orcid.org/0000-0001-7155-5044
Di Chen ⓘ http://orcid.org/0000-0002-4258-3457
Xiaochun Bai ⓘ http://orcid.org/0000-0001-9631-4781
Guozhi Xiao ⓘ http://orcid.org/0000-0002-4269-2450

## Ethics
All animal experiments were approved and conducted in the specific pathogen free (SPF) Experimental Animal Center of Southern University of Science and Technology (Approval number: 20200074).

## Decision letter and Author response
Decision letter https://doi.org/10.7554/eLife.81792.sa1
Author response https://doi.org/10.7554/eLife.81792.sa2

---

# Additional files

## Supplementary files
• Supplementary file 1. The antibody information and primer sequences.
(a) Antibody information. (b) Primer information for mouse. (c) primer information for human.
• MDAR checklist

## Data availability
All data generated or analysed during this study are included in the manuscript and supporting file; Source Data files have been provided for Figures 1-7 and figure supplements.

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
