## [Editor Report]

Gao et al. developed various genetic permutations of mouse models of kindlin-2 deficiency in the hepatocytes to decipher its role. Hepatocyte-specific loss of kindlin-2 resulted in severe inflammatory liver injury, accelerated fibrosis/portal hypertension, and massive hepatocyte cell death by apoptosis. These effects are reversed by ablation of TNF signally or by caspase 8 deletion. AAV-mediated replacement of kindlin-2 protects the mice from chemically induced acute liver injury. Together the findings are novel with significant translational potential.

---

## [Decision Letter]

**Decision letter after peer review:**

Thank you for submitting your article "Kindlin-2 inhibits TNF/NF-κB-caspase 8 pathway in hepatocytes to maintain liver development and function" for consideration by *eLife*. Your article has been reviewed by 2 peer reviewers, one of whom is a member of our Board of Reviewing Editors, and the evaluation has been overseen by Mone Zaidi as the Senior Editor. The reviewers have opted to remain anonymous.

Essential revisions:

1) The disconnect between known functions of kindlin2 in focal adhesions and cytoskeletal architecture and the reported liver dysfunction. At a minimum, I think the authors should provide some characterization of kindlin2-deficient hepatocytes and then relate this to the liver phenotype.

2) The relationship between kindlin2 loss, inflammation, and apoptosis. Specifically, does kindlin loss rapidly induce initiators of these events (TNFalpha, caspase 8), or is this secondary to a generalized destabilization of liver structure? I think the authors could do this relatively easily by comparing the time course of changes in hepatocyte structure with TNF and caspase induction.

3) Needs careful editing for typos and grammar.

4) With ascites and splenomegaly, associated with severe fibrosis, the kindlin-2 deficient mice clearly developed portal hypertension. This should be stated.

5) Massive inflammatory cell infiltrate in the livers of kindlin-2 def mice included macrophages. What other cells were present? Based on gene expression data (Figure 4), we expect to see increased B cells and neutrophils.

6) It is kindlin-2 def mice developed cholestasis (line 124). Need alkaline phosphatase or GGT and is desirable to have bile acid levels.

7) Interestingly, kindlin-2 def mice develop increased cholesterol but have low HDL cholesterol. This is very interesting. The same group has recently published in Nature Communications (vol 13, 1025, 2022) that kindlin-2 haploinsufficiency protects against fatty liver. I don't understand why this paper is not cited or discussed. The discrepancy between an apparent protective effect of haploinsufficiency in NAFLD vs fulminant liver injury/fibrosis in homozygous KO needs to be addressed.

8) Data are presented to show osteopenia in kindlin-2 def mice which is not discussed although multiple papers on bone effects of kindlin-2 def are cited. In view of the striking dyslipidemia seen in these mice, the authors should address the links between hepatic osteodystrophy and cholesterol transport (Cell Metabolism https://doi.org/10.1016/j.cmet.2022.02.007)

9) In the data on complete blood count, there is no mention of platelet counts – with portal hypertension, platelets are likely to be lower. Please provide

*Reviewer #1 (Recommendations for the authors):*

1. Needs careful editing for typos and grammar.

2. With ascites and splenomegaly, associated with severe fibrosis, the kindlin-2 deficient mice clearly developed portal hypertension. This should be stated.

3. Massive inflammatory cell infiltrate in the livers of kindlin-2 def mice included macrophages. What other cells were present? Based on gene expression data (Figure 4), we expect to see increased B cells and neutrophils.

4. It is kindlin-2 def mice developed cholestasis (line 124). Need alkaline phosphatase or GGT and is desirable to have bile acid levels.

5. Interestingly, kindlin-2 def mice develop increased cholesterol but have low HDL cholesterol. This is very interesting. The same group has recently published in Nature Communications (vol 13, 1025, 2022) that kindlin-2 haploinsufficiency protects against fatty liver. I don't understand why this paper is not cited or discussed. The discrepancy between an apparent protective effect of haploinsufficiency in NAFLD vs fulminant liver injury/fibrosis in homozygous KO needs to be addressed.

6. Data are presented to show osteopenia in kindlin-2 def mice which is not discussed although multiple papers on bone effects of kindlin-2 def are cited. In view of the striking dyslipidemia seen in these mice, the authors should address the links between hepatic osteodystrophy and cholesterol transport (Cell Metabolism https://doi.org/10.1016/j.cmet.2022.02.007)

7. In the data on complete blood count, there is no mention of platelet counts – with portal hypertension, platelets are likely to be lower. Please provide

*Reviewer #2 (Recommendations for the authors):*

1) The study provides strong evidence that kindlin-2 loss in hepatocytes leads to profound disruption of liver function and associated fibrosis. However, little attention is paid to underlying changes in liver structure which are clearly disrupted in knockouts. Given the known roles of this protein in focal adhesion organization and integrin signaling, it would be important to examine the properties of kindlin-deficient hepatocytes such as focal adhesion and cytoskeletal organization, and ECM distribution.

2) No obvious connection is made between Kindlin loss and increased inflammation and hepatocyte apoptosis. Is it possible that overall disruption of the liver structure due to focal adhesion or related disruptions leads to generalized tissue necrosis, inflammation, and cell apoptosis versus Kindlin having a more direct repressive effect on TNFalpha-mediated inflammation?

A possible insight into the relationship between these events might be gained by comparing the time course of kindlin-2 inactivation with TNF and caspase activation.

3) Figure S2 shows loss of bone volume, bone mineral density, and trabecular parameters in Fermt2 KO mice. However, no comment is made concerning the relationship between this observation and the loss of kindlin in hepatocytes, which implies an indirect effect on bone.

4) Figure 3 shows that kindlin-2 loss increases fibrosis as measured by increased collagen staining, Col1a1 mRNA, Timp1, and α SMA levels. Is this fibrosis an early or late response to kindlin loss or Is it a secondary response to increased TNF and inflammation? If hepatocytes are undergoing apoptosis, what is the source of the increased collagen?

5) The study shown in Figure 7 shows a small protective effect of Kindlin-2 overexpression on the acute liver toxicity induced by D-galactosamine/LPS treatment. The experiment only compares liver parameters between D-galactosamine/LPS treated control and kindlin overexpression mice. Does Kindlin2 overexpression affect liver parameters in the absence of D-galactosamine/LPS treatment or does it selectively protect against D-galactosamine/LPS-induced inflammation?

---

## [Author Response]

Essential revisions:1) The disconnect between known functions of kindlin2 in focal adhesions and cytoskeletal architecture and the reported liver dysfunction. At a minimum, I think the authors should provide some characterization of kindlin2-deficient hepatocytes and then relate this to the liver phenotype.

As suggested, during the revision, we have performed new experiments in Huh7 cell line. Results from IF staining showed that Kindlin-2 loss markedly altered the focal adhesion formation and impaired the cytoskeleton, cell spreading and attachment. These new results are now added as the new Figure 4o and 4p and described in the text (page 11, lines 233-236).

2) The relationship between kindlin2 loss, inflammation, and apoptosis. Specifically, does kindlin loss rapidly induce initiators of these events (TNFalpha, caspase 8), or is this secondary to a generalized destabilization of liver structure? I think the authors could do this relatively easily by comparing the time course of changes in hepatocyte structure with TNF and caspase induction.

During the revision, we have performed time-course experiments by IHC staining of livers sections of the two genotypes to measure expression of p65 and active caspase 3 proteins at 2-, 3- and 4-weeks of ages. The results showed that Kindlin-2 loss did not cause marked alterations in expression of p65 and active caspase 3 proteins and liver histology at 2-week of age. Kindlin-2 loss caused a significant elevation in expression of p65 and active caspase 3 proteins without causing marked liver injure at 3-weeks of age. At 4-weeks of age, Kindlin-2 loss caused significant up-regulation of p65 and active caspase 3 and marked structural damage in KO livers. These results clearly show that alterations in inflammation and apoptosis occur before the destabilization of liver structure in KO mice. Thus, our results support the notion that Kindlin-2 loss rapidly induces inflammation and apoptosis not due to secondary to a generalized destabilization of liver structure. These results are now added as the new Figure 2—figure supplement 1b and 1c and described in the text (page 8, lines 173-175). We thank the reviewer and editor for raising this critical issue and the suggested experiments to address it!

3) Needs careful editing for typos and grammar.

During the revision, we have tried our best to fix typos and grammar errors.

4) With ascites and splenomegaly, associated with severe fibrosis, the kindlin-2 deficient mice clearly developed portal hypertension. This should be stated.

We have added a short discussion regarding this point in the revised manuscript (page 17, line 353-355). Thank you!

5) Massive inflammatory cell infiltrate in the livers of kindlin-2 def mice included macrophages. What other cells were present? Based on gene expression data (Figure 4), we expect to see increased B cells and neutrophils.

As suggested, we have performed new experiments and found that both CD19 (B cell marker) and Ly6G (neutrophils marker) were up-regulated in KO versus control livers. These new results are now added as the Figure 4f and 4g and described in the text (page 10, lines 219-221).

6) It is kindlin-2 def mice developed cholestasis (line 124). Need alkaline phosphatase or GGT and is desirable to have bile acid levels.

Results from our new experiments showed that the levels of alkaline phosphatase, GGT and bile acid levels were increased in KO relative to those in control mice. These results are now added as the Figure 1—figure supplement 3b and described in the text (page 7, lines 146-149).

7) Interestingly, kindlin-2 def mice develop increased cholesterol but have low HDL cholesterol. This is very interesting. The same group has recently published in Nature Communications (vol 13, 1025, 2022) that kindlin-2 haploinsufficiency protects against fatty liver. I don't understand why this paper is not cited or discussed. The discrepancy between an apparent protective effect of haploinsufficiency in NAFLD vs fulminant liver injury/fibrosis in homozygous KO needs to be addressed.

The finding from our previous study is now described and cited (pages 5-6, lines 108-112). Thank you!

8) Data are presented to show osteopenia in kindlin-2 def mice which is not discussed although multiple papers on bone effects of kindlin-2 def are cited. In view of the striking dyslipidemia seen in these mice, the authors should address the links between hepatic osteodystrophy and cholesterol transport (Cell Metabolism https://doi.org/10.1016/j.cmet.2022.02.007)

As suggested, a short discussion regarding the raised point (page 17, lines 355-358) and the reference are now added. Thank you!

9) In the data on complete blood count, there is no mention of platelet counts – with portal hypertension, platelets are likely to be lower. Please provide

Our new results showed that the platelet number was reduced in KO versus control mice. These results are now added as the new Figure 4—figure supplement 1 and described in the text (page 11, lines 222-224).

Reviewer #2 (Recommendations for the authors):1) The study provides strong evidence that kindlin-2 loss in hepatocytes leads to profound disruption of liver function and associated fibrosis. However, little attention is paid to underlying changes in liver structure which are clearly disrupted in knockouts. Given the known roles of this protein in focal adhesion organization and integrin signaling, it would be important to examine the properties of kindlin-deficient hepatocytes such as focal adhesion and cytoskeletal organization, and ECM distribution.

Please see our response to above point 1.

2) No obvious connection is made between Kindlin loss and increased inflammation and hepatocyte apoptosis. Is it possible that overall disruption of the liver structure due to focal adhesion or related disruptions leads to generalized tissue necrosis, inflammation, and cell apoptosis versus Kindlin having a more direct repressive effect on TNFalpha-mediated inflammation?A possible insight into the relationship between these events might be gained by comparing the time course of kindlin-2 inactivation with TNF and caspase activation.

Please see our response to above point 2.

3) Figure S2 shows loss of bone volume, bone mineral density, and trabecular parameters in Fermt2 KO mice. However, no comment is made concerning the relationship between this observation and the loss of kindlin in hepatocytes, which implies an indirect effect on bone.

Please see our response to above point 8.

4) Figure 3 shows that kindlin-2 loss increases fibrosis as measured by increased collagen staining, Col1a1 mRNA, Timp1, and α SMA levels. Is this fibrosis an early or late response to kindlin loss or is it a secondary response to increased TNF and inflammation? If hepatocytes are undergoing apoptosis, what is the source of the increased collagen?

The enhanced fibrosis in KO liver is probably a late response to Kindlin-2 loss. Along with hepatocyte apoptosis, a large number of nonparenchymal cells, including macrophages (Figure 4e), astrocytes (Figure 2i) and cholangiocytes (Figure 2h), were observed in KO liver. This may contribute to the increased collagen accumulation in KO liver. We have discussed this in the revised manuscript (page 16, lines 350-352).

5) The study shown in Figure 7 shows a small protective effect of Kindlin-2 overexpression on the acute liver toxicity induced by D-galactosamine/LPS treatment. The experiment only compares liver parameters between D-galactosamine/LPS treated control and kindlin overexpression mice. Does Kindlin2 overexpression affect liver parameters in the absence of D-galactosamine/LPS treatment or does it selectively protect against D-galactosamine/LPS-induced inflammation?

Our results showed that Kindlin-2 overexpression did not markedly alter the liver function without the D-galactosamine/LPS treatment. These new results are now added in the new Figure 7—figure supplement 1 and described in the text (page 14, lines 295-296).